# Characterization of the Bacterial Composition of 47 Fermented Foods in Sweden

**DOI:** 10.3390/foods12203827

**Published:** 2023-10-19

**Authors:** Marie Palmnäs-Bédard, Aline de Santa Izabel, Johan Dicksved, Rikard Landberg

**Affiliations:** 1Department of Life Sciences, Division of Food and Nutrition Science, Chalmers University of Technology, 412 96 Gothenburg, Sweden; rikard.landberg@chalmers.se; 2YOGUT ME AB, 170 70 Solna, Sweden; aline@synbiotickitchen.com; 3Department of Animal Nutrition and Management, Swedish University of Agricultural Sciences, 750 07 Uppsala, Sweden; johan.dicksved@slu.se; 4Department of Public Health and Clinical Medicine, Umeå University, 901 87 Umeå, Sweden

**Keywords:** fermented foods, fermented beverages, microbiota, bacterial composition

## Abstract

Fermentation has long been utilized to preserve and enhance the flavor and nutritional value of foods. Recently, fermented foods have gained popularity, reaching new consumer groups due to perceived health benefits. However, the microbial composition of many fermented foods re-mains unknown. Here, we characterized the bacterial composition, diversity, and richness of 47 fermented foods available in Sweden, including kombucha, water kefir, milk kefir, yogurt, plant-based yogurt alternatives, kimchi, sauerkraut, and fermented vegetables. Via 16S rRNA gene sequencing, we identified 2497 bacteria (amplicon sequence variants). The bacterial composition was strongly associated with the type of fermented food, and lactic acid bacteria and/or acetic acid bacteria dominated most samples. However, each fermented food had a unique composition, with kombucha and water kefir having the highest diversity across and within samples. Few bacteria were abundant in multiple foods and food groups. These were *Streptococcus thermophilus* in yogurts and plant-based yoghurts; *Lactococcus lactis* in milk kefirs and one water kefir; and *Lactiplantibacillus plantarum* in kimchi, sauerkraut, and fermented cucumber. The broad range of fermented foods included in this study and their diverse bacterial communities warrant further investigation into the implications of microbial compositions for product traits and potential impact on human health.

## 1. Introduction

Fermented foods represent a diverse group of foods defined as “foods made through desired microbial growth and enzymatic conversion of food components” [1]. During fermentation, substrates present in the raw materials are biochemically modified, which may lead to increased food stability and safety as well as improved nutritional value and sensory properties [1]. Humans have consumed fermented foods for thousands of years, and fermented foods remain an important part of many diets worldwide, with many foods being unique to geographical areas and communities [2,3]. More recently, fermented foods have become increasingly popular, and a broad range of both traditionally made and industrially produced fermented foods are now available to consumers. This results from a raised interest in fermented foods and their perceived health benefits, an increased utilization of fermented ingredients in the food industry, and the need for food preservation [4,5,6]. However, despite the ubiquitous consumption of fermented foods and the projected continued growth of the fermented food market, there is still a lack of knowledge regarding the microbial community (microbiota) of many fermented foods, limiting the understanding of its role on specific product traits and potential effects on human health.

Recent development and application of sequencing technologies has complemented traditional microbiology approaches and led to the identification of both culturable and unculturable microbes in fermented foods, and the characterization of large and complex microbial communities that vary in taxonomy, function, and abundance [1,6]. Lactic acid bacteria (LAB) dominate in many fermented foods, while other foods contain high proportions of other bacteria, e.g., acetic acid bacteria (AAB), as well as fungi and archaea [1]. Bacterial communities in dairy products and fermented foods from Asia have most commonly been investigated [7,8,9,10]. Studies that characterize and compare the microbiota of fermented food products are also emerging and include different dairy kefirs [10,11], kimchi [7,12], carrot juice fermentations [13], and meat products [14] as well as a broad range of traditional African, Asian, and European fermented foods [15,16,17]. However, comparisons for other types of fermented foods are still lacking, especially for foods available for consumption in Scandinavian countries. Importantly, as the microbiota of fermented foods depends on the starting ingredients, initial microbial community, and mode of fermentation (i.e., whether spontaneous or initiated via inoculation or starting cultures) as well as on production processes and environment [3], there may be distinct differences across foods, depending on geographic location, manufacturer, batch, cooking environment, time of consumption, and whether fermented foods are made by individuals or produced on an industrial scale.

In the present study, we aimed to investigate the bacterial composition, diversity, and richness of a broad range of fermented foods that were available on the Swedish market or prepared in home environments in Sweden. Both home-made and commercial products were selected to represent different types of fermented products available to the population and to allow for comparisons between home-made and commercially produced foods. The foods belong to eight different fermented food groups, namely, kombucha, water kefir (also known as tibicos), milk kefir, yogurt, plant-based alternatives to yogurt, sauerkraut, kimchi, and various fermented vegetables, including fermented beetroot and cabbage, cucumber, and carrot.

## 2. Materials and Methods

### 2.1. Collection of Fermented Foods

In this study, we collected 49 fermented foods, but only 47 were included in the statis-tical analysis due to the exclusion of two samples during microbiota data preparation, as described below (Section 2.4). Fermented foods were either purchased from local grocery stores and health stores in Sweden or collected by two individuals making fermented foods at their own leisure in their homes over a two-month period (Stockholm, Sweden from January to February 2020) (Table 1). In response to this practical approach, the collection of home-made samples is broad and heterogeneous. Starter cultures for yoghurts and plant-based alternatives to yoghurts, and so-called water and milk kefir “grains”, i.e., symbiotic cultures of bacteria and yeast (SCOBY), were bought online or from a local health store (Gryningen, Stockholm, Sweden). Information on the bacterial composition of the starter cultures was extracted when available, either from the product information on the packaging, from the website where the product was purchased, or as provided by company representatives upon request (Table 1). There was no or incomplete information regarding the bacterial composition of the starter cultures used to produce commercial yoghurts and plant-based alternatives to yoghurts (Table 1).

### 2.2. Preparing Fermented Foods in Home Environments

Sauerkrauts were prepared by two individuals using white cabbage from different grocery stores and variations of the same recipe (Table 1). In brief, the outer leaves of the cabbage were discarded prior to chopping and mixing the vegetables. Salt and spices were added and the mixture massaged until a brine formed; the mixture was then placed in a jar, ensuring that vegetables were submerged in the brine. The jars were first left to ferment at room temperature, then placed in a refrigerator. Kimchi was prepared by one individual using the same type of white cabbage as for most sauerkrauts and a dry-salting fermentation method. This included chopping the cabbage; adding salt, spices, and green onion; and massaging the vegetables. The excess liquid was strained prior to packing the vegetables in the jar and leaving the jar to ferment at room temperature. Kombucha was prepared by one individual using a combination of teas, sugar, and a kombucha SCOBY belonging to the individual. Sugar was dissolved in boiled water, followed by adding and brewing the tea and letting the tea cool down to room temperature prior to adding the SCOBY. The fermentation period lasted 11 days, with the second and third batch inoculated from the preceding batch. Milk kefirs, water kefirs, yoghurts, and plant-based alternatives to yoghurts were prepared by one individual in their home. All milk kefirs were prepared using the same type and brand of pasteurized milk and the same type of milk kefir grains, with the exception of one sample, which used a commercial starter culture. Milk and milk kefir grains were added to a jar and stirred; the jar was then covered with a tight-weave cloth secured with a rubber band and left to ferment at room temperature. Fermentation lasted three days but with different starting dates during the same month (January 2020). Once fermentation was completed, the liquid and grains were separated using a mesh strainer. The two water kefirs were prepared using the same ingredients (water, sugar, and water kefir grains) and recipe, but one batch fermented for three days whilst the other fermented for four days. Water, water kefir grains, and sugar were added to a jar and stirred until most of the sugar had dissolved, then covered with a tight-weave cloth secured with a rubber band. After fermentation, the liquid was separated from the grains using a fine mesh strainer. All yoghurts were prepared on the same day using the same brand and type of pasteurized milk, but with different starter cultures (Table 1). The milk was first brought to a boil, then allowed to cool to 42 °C prior to the addition of starter cultures, as measured using a manual digital infrared food thermometer. A kitchen appliance yogurt maker was used to maintain a temperature between 42 and 45 °C during fermentation. Plant-based alternatives to yoghurt were made using the same procedures, except made with coconut milk. Bacterial cultures and starting dates varied, and one plant-based alternative to yoghurt contained additional ingredients, namely, cashew nuts and tapioca starch. Individuals used clean equipment and glass jars and washed their hands with soap and water prior to preparing the fermented foods.

### 2.3. Sample Preparation and Handling

Fermented foods were stored in refrigerators in home environments after preparation or purchase (unopened containers) prior to sampling. One representative sample was taken from each of the 49 foods; these were placed in separate 15 mL falcon centrifuge tubes (Sarstedt, Nümbrecht, Germany) and immediately placed in the freezer. Hands were washed with water and soap prior to sampling. Solid foods were sampled using sterile disposable spoons and by sampling different spatial locations of the food. Liquids were stirred with sterile disposable spoons, then poured. Each sterile spoon was only used for one sample. Within 30 days after freezing, all samples were transported on ice for about 1.5 h on one occasion. Samples remained frozen and were stored at −80 °C prior to microbial analysis.

### 2.4. Microbiota Analysis

Total DNA was extracted from 4 mL of sample aliquots of each fermented food. Prior to collection of the aliquots, falcon tubes containing the fermented samples were thawed and thoroughly homogenized via vortex mixing. Subsequently, two 2 mL aliquots were transferred to two 2 mL sterile Eppendorf tubes. The aliquots were centrifuged for 5 min at 13,000 rpm to pellet particles. The supernatant was discarded and the remaining pellet in each tube was resuspended in 0.5 mL InhibitEX buffer. Replicate resuspensions from the same falcon tubes were then pooled, and DNA was isolated from the samples using a QIAamp Fast DNA Stool mini kit (Qiagen, Hilden, Germany) according to the protocol from the manufacturer, except that the bacterial cell walls were mechanically disrupted with 0.1 mm Zirconium/Silica beads (BioSpec products, Bartlesville, OK, USA) 2 × 60 s, using a Precellys Evolution (Bertin Technologies, Mon-tigny-le-Bretonneaux, France). Amplicons from the V3 and V4 regions of the 16S ribosomal RNA gene were generated from the extracted DNA using the primers 341F and 805R. For the polymerase chain reactions (PCRs), Phusion^®^ High-Fidelity PCR Master Mix kits (New England Biolabs, Ipswich, MA, USA) were used, and the PCR products were purified with Qiagen Gel Extraction Kits (Qiagen, Hilden, Germany) and quantified with a Qubit^®^3.0 Fluorometer. The final libraries were generated with NEBNext^®^ UltraTM DNA Library Prep Kits (New England Biolabs) that incorporated barcodes and adaptors. The amplicons were then sequenced using the Illumina Novaseq 6000 (2 × 250 bp) platform provided by Novogene, Beijing, China. One sample that did not pass the initial quality control was not included in the sequence analysis.

The raw demultiplexed reads from the sequencing were processed using the DADA2 pipeline to denoise (with the following parameters used in the filterAndTrim-step: maxN = 0, maxEE = c(2,2), truncQ = 2, rm.phix = TRUE, compress = TRUE), dereplicate reads, merge pair end reads, and remove chimeras [18]. The table of amplicon sequence variants (ASVs) were assigned to reference sequences using the naive Bayesian classifier called with the assignTaxonomy command [19] against the SILVA rRNA database [20], release 138, formatted for DADA2 by B. Callahan (https://benjjneb.github.io/dada2/training.html, accessed on 5 January 2021). This included assignments applying the former *Lactobacillus* classification, which since has been re-classified into 25 genera [21] and synonyms for phyla such as Bacteroidetes (Bacteroidota) and Campylobacteriota (listed as Campilobacterota). After quality filtration of the sequence data, the final dataset contained 4.7 M sequences, with an average of 100.6 ± 16.1 thousand sequences per sample. One sample failed in the sequence analysis and generated few sequences. Thus, this sample was excluded from further analyses. A total of 2535 ASVs were identified. ASVs classified as mitochondria on family level (N = 12) or chloroplast on order level (N = 26) were removed, and thus a final total of 2497 ASVs were included in the statistical analysis.

### 2.5. Taxonomic Assignment for Individual ASV with High Relative Abundance

Taxonomic assignments were performed using the ASV sequence and the default pa-rameters in NCBI Nucleotide blast suite (https://blast.ncbi.nlm.nih.gov/, accessed on 27 May 2022) for ASVs present at ≥1% in at least one sample (N = 143) (Table S1). Notably, for species level assignments, the accuracy cannot be assured; rather, the best match is reported, whilst the genus level is generally considered reliable. In the manuscript, taxa were reported using the name recognized by NCBI [22] at time of submission, with exception of ASVs, for which the best species match was <80% of identified hits. In this case, upstream taxonomic levels that fulfilled the 80% criteria were used, most commonly the genus level or family level.

### 2.6. Statistical Analysis

All statistical analysis was performed in the R environment. The ASV data table, tax-onomic information, and sample data (i.e., type of fermented food and commercial or made in home environment) were first converted and saved as a phyloseq object using the readxl, tibble, and phyloseq packages. Alpha diversity (Shannon and Inverted Simpson diversity) and richness (observed richness and Chao1) were calculated using Phyloseq after normalizing the ASV data using the scaling with ranked subsampling method (SRS package) [23]. Results were summarized using dplyr version 1.1.1. and presented as box plots. Differences in alpha diversity and richness across all groups of fermented foods were determined via Kruskal–Wallis rank sum test. Pairwise comparisons using the Wilcoxon rank sum test were then used to identify all statistically significant differences in alpha diversity, and richness between groups and *p*-values were adjusted using the Benjamini–Hochberg method. The relative abundance of ASVs was calculated using phyloseq (Appendix A) and visualized as bar plots. Mean and standard deviation were calculated on the phylum and genus levels after aggregating ASVs based on taxonomy using the aggregate function of the phyloseq package. Dissimilarities in bacterial composition were assessed by conducting principal coordinate analysis (PCoA) and bi-plot visualization based on un-scaled ASV data on relative abundance and Bray–Curtis distance using the vegan and Ecodist packages. For visual interpretation purposes, ordination was performed using only the ASVs having a relative abundance of ≥1% in at least one sample. Permanova (permutational multivariate analysis of variance) was then performed on the Bray–Curtis distance matrix to compare differences across all fermented food categories as well as pairwise comparisons of food categories. The permanova was conducted with the adonis function of the vegan package, with 10,000 permutations and with *p*-value correction using the Benjamini–Hochberg method. A heatmap for highly abundant ASVs was generated using the heatmap.2 function and vegan package version 2.6-4, clustering samples and ASVs using a Bray–Curtis dissimilarity matrix and average linkage based on the relative abundance data for all ASVs that were present in quantities ≥10% in at least one sample.

The relative abundance of genera commonly classified as LAB (N = 8) [24] or AAB (N = 4) [25] (Appendix A), respectively, were aggregated to represent a crude estimate of the total proportion of LAB and AAB in each food and food group. Differential composition of estimated LAB and AAB across fermented food groups were determined via Kruskal–Wallis rank sum test followed by Wilcoxon rank sum test correcting for multiple comparisons. A *p*-value, or a *p*-value corrected for multiple comparisons when applicable, of <0.05 was considered statistically significant. Figures were generated using the ggplot2 package.

## 3. Results and Discussion

The microbiota of 47 out of the 49 fermented foods were included in the analysis and are discussed below (Table 1). Excluded products were a fermented “shot” beverage, for which DNA isolation was unsuccessful and the bacterial content could not be measured, and a commercial kombucha that failed to generate data in the sequence analysis. In total, 2497 ASVs belonging to 33 phyla and 386 genera were observed (Appendix A).

### 3.1. Alpha-Diversity and Richness Differed by Fermented Food Group

Alpha diversity and richness were significantly different across fermented food groups, indicating differences in the community structure (number and proportions) of bacteria responsible for the fermentation and bacteria otherwise present in the foods (Figure 1). Kombucha had the highest Shannon diversity and richness, and estimates were particularly high for two home-made and one commercial kombucha as well as for one commercial water kefir (Figure 1, Appendix A). These fermented beverages were made using symbiotic cultures of several bacteria and yeast, likely varying in their composition. Shannon diversity and richness were lowest for plant-based alternatives to yoghurt, followed by yoghurts or various fermented vegetables (Figure 1, Appendix A). The low diversity of yoghurts and plant-based alternatives to yoghurt are aligned with typical production practices, i.e., using a small and defined selection of starter cultures (Table 1). Notably, some yogurts were also supplemented with probiotic bacteria, which contribute to the measured diversity and richness (Table 1). No differences were found for inverse Simpson estimates of alpha diversity after correcting for multiple comparisons (Appendix A).

### 3.2. Dissimilarities in the Bacterial Community across Fermented Foods and Food Groups

The bacterial composition of the fermented foods differed according to fermented food group (*p* < 0.001) (Figure 2a, Appendix A). Yogurts and plant-based alternatives to yoghurts were distinct from other fermented foods, and fermented beverages (kombucha and water kefir) and milk kefir clustered separately from fermented foods (sauerkraut, kimchi, various vegetables, yoghurt, and plant-based alternatives to yoghurt) (Figure 2a). Some samples resembled another fermented food group; a home-made kimchi, made with white cabbage rather than savoy cabbage, was more similar to sauerkrauts than kimchi, whilst a fermented cucumber product resembled kimchi and a commercial water kefir clustered amongst milk kefirs.

Bacteria belonging to Firmicutes, a phylum containing most LAB, were present in all fermented food groups, whilst variation within the Proteobacteria phylum contributed to differences across food groups and were absent in yoghurts and plant-based alternatives to yoghurts (Figure 2b). Bacterioidetes, Actinobacteria, and Campylobacterota were highly present only in select foods and food groups. Notably, some fermented foods, such as kombucha, water kefir, and milk kefir, also contain yeasts [4,8,11,26,27], which were not measured in the present study.

### 3.3. The Bacterial Composition of the Fermented Foods and Food Groups

Bacteria belonging to the four phyla Firmicutes, Proteobacteria, Actinobacteria, and Bacteroidetes comprised ≥95% of the total abundance for each food and food group, yet in varying proportions (Figure 3a, Appendix A). These phyla comprise LAB, acetic acid bacteria, common probiotic bacteria (e.g., bifidobacteria), and potential contaminants (e.g., from soil or during preparation of the fermented food). Most ASVs had low (<1.0%, N = 2354) or very low (<0.1%, N = 1915) abundance in all samples, together contributing to 1.9 ± 3.3% and 9.3% ± 13.5, respectively, of the total abundance in the fermented foods (Figure 3b, Appendix A). Kombucha and water kefir contained the highest proportions of ASVs that hadlow and very low abundance, in agreement with the higher alpha diversity and richness. Low and very low abundance taxa accounted for 60.7% and 1.8%, respectively, of the abundance for kombucha and 34.7% and 6.3%, respectively, for water kefir. ASVs with high relative abundance (≥10% in at least one food, N = 37 ASVs) were predominantly LAB (*Lactobacillus*, *Lactococcus*, *Leuconostoc*, *Pediococcus*, *Schleiferilactobacillus*, and *Streptococcus* and genera formerly classified as *Lactobacillus* [21], i.e., *Lactiplantibacillus*, *Latilactobacillus*, *Lentilactobacillus*, *Levilactobacillus*, and *Loigolactobacillus*), AAB (*Acetobacter*, *Acetobacteraceae* species, and *Gluconobacter*) or *Bifidobacteria* and in rare cases ethanol-producing bacteria (*Zymomonas*), *Bacillaceae*, or *Enterobacter* (contaminant) (Figure 3b and Figure 4, Appendix A).

The estimated total proportion of LAB and AAB showed LAB to predominate in kimchi, various vegetables, yoghurt, and most sauerkrauts and to be highly abundant in milk kefirs and plant-based alternatives to yoghurt (Appendix A, Appendix A). AAB were distinct for kombucha, milk kefir, and water kefir, albeit with considerable variation across samples (Appendix A, Table 1).

### 3.4. Bacterial Composition and Variability within Each Fermented Food Group

#### 3.4.1. Kombucha

The diversity across the five kombucha samples was high. The proportion of LAB, AAB, and other bacteria varied greatly, ranging from 1 to 90% estimated AAB and 2 to 20% estimated LAB (Figure 4 and Appendix A, Appendix A). Most taxa were present in low proportions, and many were classified as uncultured bacteria (Appendix A). Highly abundant bacteria were predominately AAB, e.g., *Gluconobacter* spp. and *Acetobacteraceae* spp., either dominating in a commercial kombucha (Sample 20) or in two home-made kombuchas (Samples 1 and 2) (Figure 3b and Figure 4). Interestingly, these ASVs were nearly absent in the remaining kombucha (Samples 21 and 3), which were instead characterized by ASVs with relatively low abundance, mainly a combination of LAB and unknown bacteria. *Lactobacillus sakei* and *Lactococcus lactis* were the most abundant LAB, present at the most at 7.7% and 3.4%, respectively, in home-made kombucha Sample 3. The high proportions of AAB and unclassified bacteria are in accordance with previous studies on kombucha, yet there is considerable diversity both across and within studies in terms of the identity of dominating microbiota [28,29,30,31,32].

We found no clear distinction between commercial and home-made kombucha, or resemblance among home-made kombucha, although the home-made kombuchas were made consecutively using the same SCOBY and recipe. Instead, several ASVs with lower abundance, including several LAB, increased gradually from the first to the third and final batch, whilst *Gluconobacter* sp. and an unknown *Acetobacteraceae* species decreased gradually and considerably, from 30.8% and 42.1% to 0.4 and 0.2%, respectively.

#### 3.4.2. Water Kefir

Home-made water kefirs and commercial Sample 30 were characterized by higher AAB, whereas remaining commercial water kefirs had higher proportions of LAB (Figure 3b, Figure 4 and Appendix A, Appendix A). Each water kefir was dominated by a unique ASV, alone comprising 22.2–57.7% of the total abundance in each sample. Few taxa were present at relatively high proportions, and many could not be assigned to a putative species or were classified as uncultured bacteria.

Two home-made water kefirs, prepared using the same recipe, batch of water kefir grains, and ingredients but with different fermentation times, showed relatively strong resemblance and were characterized by a high abundance of *Acetobacter* sp., *Enterobacter cloacae* complex, and *Gluconobacter* sp. Proportions of the *E. cloacae* complex, a likely contaminant, were considerably lower in the water kefir that had fermented for 4 days (Sample 27) compared to that which had fermented for 3 days (Sample 26), while the estimated proportion of LAB and AAB increased twofold. Water kefirs from the same company (Samples 30 and 31) showed limited similarity and differed in terms of flavour ingredient (rhubarb juice vs. rosehip). Sample 30 had high proportions of an unknown Enterobacterales bacteria and several *Gluconobacter*. In contrast, Sample 31 had the highest diversity amongst water kefirs and was dominated by *L. lactis*, the latter similar to milk kefir (see below in Section 3.4.3), explaining at least partly why sample 31 had shown higher overall resemblance to milk kefirs compared to water kefirs (Figure 2). Commercial water kefir Sample 29 was dominated by *Zymomonas mobilis*, an ethanol-producing bacteria previously found to dominate in water kefir from three different countries [10]. Our results showed some consistency with the starter culture used to make Sample 28 (Table 1) and with reported literature in terms of the high and variable proportions of LAB and AAB. However, Sample 28 was dominated by a bacterium classified as *Niallia nealsonii*, belonging to the family *Bacillaceae*, not reported in the starter culture and thus of unknown origin (Table 1). Production processes, in addition to the composition of the water kefir grains and choice of ingredients, may differentially favour some microbes; for example, higher abundances of AAB have been found in aerobic and low-nutrient conditions and higher presence of yeast and/or ethanol [27,33], whilst the composition of LAB may depend on back-sloping procedures and temperature [34].

#### 3.4.3. Milk Kefir

Milk kefirs contained few highly abundant bacteria. Among these, *L. lactis* was highly abundant in all milk kefirs, in alignment with several other studies [35,36]. *L. lactis* was particularly dominant in commercial milk kefirs and home-made Sample 25, constituting 75.1–87.7% of the total abundance vs. 14.4–22.0% in the other home-made milk kefir (Figure 4, Appendix A). In fact, there was a clear distinction between the home-made Samples 8–10, fermented using milk kefir grains (with an unknown microbial composition), and the commercial milk kefirs and Sample 25, made using selected yet unspecified starter cultures (Table 1). From the list of bacteria supplemented to the commercial milk kefirs, only *B. animalis* was found to be present at ≥1% (1.7% in Sample 24). Home-made milk kefir Samples 8–10 had high proportions of both LAB and AAB taxa, contributing at the most to 39.8–52.4% of the total abundance whilst the second most abundant ASVs contributed with 18.5–28.2% to each sample (Figure 3b). Several of these were unique to home-made Samples 8–10, namely *Lentilactobacillus* sp., possibly *L. kefiri*, a core bacterium for milk kefirs and known producer of the polysaccharides constituting milk kefir grains [35,36], and *Acetobacter* sp. as well as some bacteria present in the milk kefir grains, including *L. lactis* and *Leuconostoc* spp., but not cultures of *Streptococcus* and *Levilactobacillus*. For commercial milk kefirs and Sample 25, the second most abundant ASV contributed <5% of the total abundance and included *Lactiplantibacillus* sp., *Bifidobacterium animalis*, an unknown *Streptococcaceae* species, and *E. cloacae* complex, depending on sample. Out of the 12 ASVs that had an abundance ≥1.0% in at least one milk kefir, about half were absent in two or more samples.

#### 3.4.4. Yoghurt

All yoghurts were dominated by one *Streptococcus thermophilus* taxa, comprising 44.9–83.6% of the total abundance in each sample (Figure 4, Appendix A). Additionally, three other ASVs classified as *S. thermophilus* or *Streptococcus* sp. were highly abundant (5.0–30.0%), and two home-made yoghurts also contained high proportions of a bacterium classified as *Lactobacillus delbrueckii* subsp. *bulgaricus* (11.6–34.3%). This is in accordance with the known composition of the starter culture of the home-made yoghurt Sample 36 and common yoghurt production processes [37]. In this regard, the absence of *Lactobacillus delbrueckii* subsp. *Bulgaricus* in commercial yoghurts was unexpected and lacks explanation. The composition of the starter cultures for commercial yogurts were unspecified or partially provided. Remaining ASVs had low abundance (≤5%) and included *Lentilactobacillus buchneri*, *Schleiferilactobacillus harbinensis*, *Lactiplantibacillus* sp., *Gluconobacter*, *L. lactis*, and unknown bacteria.

#### 3.4.5. Plant-Based Alternatives to Yoghurt

Plant-based alternatives to yoghurt were dominated by *S. thermophilus* and/or *B. animalis*, as well as *L. delbrueckii* subsp. *bulgaricus* in commercial Sample 43 and to lesser extent *L. plantarum* and *Bacillus* spp. in home-made Sample 42 (Figure 4, Appendix A). The exception was a home-made plant-based yoghurt alternative (Sample 48) with high proportions of *Schleiferilactobacillus harbinensis* spp. and *Schleiferilactobacillus* sp. also containing several unique ASVs, including *Lentilactobacillus* spp. and unculturable bacteria, whilst lacking *B. animalis* and *Bacillus* spp. Commercial plant-based yoghurt alternatives showed particularly high resemblance to one another and were dominated by *S. thermophilus* (50.1–88.0%) and *B. animalis* (9.2–40.6%), likely due to the composition of the starter cultures. The bacterial composition of Sample 40 corresponded well to the listed starter culture, and there was partial resemblance (*B. animalis*) for Sample 39 in this regard. For unknown reasons, we did not detect *L. acidophilus* in Sample 39 or in other foods where this bacterium was listed among the starter cultures (i.e., water kefir Sample 28, milk kefir Sample 24, and yogurt Sample 32 and 36). Two home-made samples (Samples 41 and 42) were also dominated by *B. animalis* (61.9–81.1%). Although yoghurt alternatives can be produced from many plant-based sources, as represented by the samples in this study (i.e., almond, soya, and coconut), many seem to be made using similar starter cultures [38].

#### 3.4.6. Sauerkraut

For sauerkrauts, the bacterial composition was predominated by a few highly abundant LAB (Figure 3b, Figure 4 and Appendix A, Appendix A). Three home-made sauerkrauts primarily contained *L. sakei*, and Sample 7, made with red cabbage instead of white cabbage, also contained *Latilactobacillus curvatus*, *Leuconostoc mesenteroides*, and *Lactiplantibacillus* spp. A fourth home-made sauerkraut (Sample 4) showed little resemblance to other sauerkrauts and had high proportions of *Enterobacter* spp. and *Klebsiella* taxa but limited LAB, possibly resulting from contamination and a shorter fermentation time [39] (Appendix A). The three commercial sauerkrauts presented some variability; two were dominated by different *Lactiplantibacillus* taxa and had relatively high proportions of *L. brevis* and either *Leuconostoc* sp. (Sample 11) or *Pediococcus* sp. and *L. curvatus* (Sample 13). The third commercial sauerkraut (Sample 12) had high proportions of *L. mesenteroides* and *L. curvatus* spp. The bacterial composition and the dominance of LAB show high resemblance to prior studies, except that high proportions of *L. sakei* and contaminants have not been as evident previously [40,41,42].

#### 3.4.7. Kimchi

Commercial kimchi had several common characteristics, including high proportions of *Lactiplantibacillus* sp. (1.8–52.4%), *L. brevis* (1.8–51.6%), and to some extent also *L. sakei* (0.5–10.5%) (Figure 4, Appendix A). The resemblance was otherwise moderate when comparing four kimchis from the same company that contained the same ingredients, also when comparing the two “mild” (Samples 17 and 18) and “spicy” kimchis (Samples 14 and 15), representing samples of the same products but from different jars. Disparities included higher proportions of several *Lactiplantibacillus* taxa in Sample 15 and relatively high abundance of *Pediococcus* sp. and *Levilactobacillus* sp. in Sample 18. The fourth commercial kimchi from another company was dominated by *L. brevis* and partly also *Lactiplantibacillus* sp. and *L. curvatus*. The only home-made kimchi had high proportions of *Leuconostoc mesenteroides*, contributing to 81.5% of the total abundance. Our results are in high agreement with previous characterizations of the kimchi microbiota and fermentation process [43], where temperature, length of fermentation, and the microbiota of the chosen ingredients have been identified as key factors contributing to variations in the bacterial community [43]. However, we did not detect high proportion of *Weisellaor* bacteria that were not LAB, as recently described by others [12].

#### 3.4.8. Various Fermented Vegetables

The group of various fermented vegetables had relatively high proportions of different LAB, including several *Lentilactobacillus* taxa in Samples 44, 45 and 47, which was unique to these fermented foods (Figure 3b, Figure 4 and Appendix A, Appendix A). In fact, Sample 47, a fermented beetroot and cabbage product, almost exclusively contained *Lentilactobacillus* species and was dominated by *Lentilactobacillus buchneri*, similarly to a fermented cucumber product (Sample 47). This fermented cucumber product also contained *Loigolactobacillus coryniformis* and *Schleiferilactobacillus harbinensis* whereas another fermented cucumber product (Sample 46) had high proportions of *Lactiplantibacillus* sp., including *L. plantarum* and *L. sakei*, showing higher resemblance to the literature on other fermented cucumber products, also reporting high proportions of e.g., *Pediococcus* [44,45]. These three samples, all from the same company, were also characterized by having one taxon that made up about 45% of the total abundance and where an additional 41.1–49.9% of the abundance were attributed to 5–10 taxa present ≥1% in their respective sample. Interestingly, despite representing different products, these three samples showed high resemblance (Figure 3b and Figure 4). The fourth product, containing fermented carrot (Sample 45), had high proportions of an unknown bacteria (69.6%), potentially a species belonging to the *Lactobacillaceae* family and to lesser extent *Lentilactobacillus* spp. and *Pediococcus* spp. In reference to a study comparing close to 40 samples of fermented carrot juice, there was little resemblance other than a high proportion of LAB [13].

### 3.5. Final Remarks

The samples included in this study were selected using a practical approach, namely purchasing fermented foods from local grocery stores and health stores and gathering samples from individuals making fermented foods at their own leisure in their homes. Systematic investigations addressing narrower research questions can complement our study, e.g., in terms of specific product traits or factors affecting microbial composition. Due to the lower resolution of 16S rRNA sequencing in comparison to, e.g., metagenomic sequencing, strain level detection was not possible and some ASVs were matched at the genus and family level. Furthermore, we did not investigate microbial functions, for which metagenomics sequencing is better suited than 16S rRNA sequencing. These limitations may have masked additional differences and similarities across samples. Sequencing methods are also limited in the regard that it is not possible to distinguish living from dead microbes, which could be a reason for why we detected some unexpected or uncommon bacteria in the fermented foods, e.g., common intestinal bacteria. While it is not possible (and sometimes not preferable as in the case of spontaneous fermentation) to completely avoid contamination from the environment (raw materials, equipment, and environment [39]), proper hand hygiene and cleanliness of the workspace and utensils can reduce some unwanted contamination [3]. Whilst generally considered safe to consume, fermented foods are not exempt from food-borne pathogens, and we measured high proportions of intestinal bacteria (Enterobacter and Klebsiella) in a home-made sauerkraut. Lastly, the composition of the kombucha SCOBYs and the milk and water kefir grains used in this study remain unknown. Sequencing these cultures may have provided partial insight to differences across samples as the microbial composition of the kombuchas and kefirs depend partly on the kombucha SCOBYs and kefir grains [27,46]. It is also noteworthy that the home-made kombucha samples, made sequentially using the same SCOBY and same ingredients by the same individual in their home, resulted in beverages with diverse bacterial communities. This may suggest a sensitivity to changes within the environment and, subsequently, that such differences impact the microbial metabolite concentrations in kombucha and potentially differential effects on health.

## 4. Conclusions

This study characterized and compared the bacterial composition, diversity, and richness of a broad selection of fermented foods available on the Swedish market and prepared in home environments. We found the bacterial composition to vary depending on the type of fermented food, i.e., kombucha, water kefir, milk kefir, yogurt, plant-based alternatives to yogurts, kimchi, various fermented vegetables, or sauerkraut. However, the fermented foods had unique compositions, and the kombucha and the water kefir were particularly diverse across and within samples. There were also differences between home-made fermented foods and commercial products, as was most evident for milk kefirs but also plant-based alternatives to yoghurts and sauerkraut, and to some extent differences linked to the use of starter cultures in comparison to living symbiotic colonies (e.g., milk kefir grains). Only a few bacteria were highly abundant in multiple foods and fermented food groups, including *S. thermophiles*, which was highly prevalent in yoghurts and plant-based alternatives to yoghurts; *L. lactis* highly abundant in milk kefirs and one water kefir; and *L. plantarum*, characteristic of kimchi and with high abundance in one sauerkraut and a fermented cucumber product. The broad range of fermented foods included in this work and their diverse bacterial communities warrant further investigations into the implications of microbial compositions for product traits and potential impact on human health.

## Figures and Tables

**Figure 1 foods-12-03827-f001:**
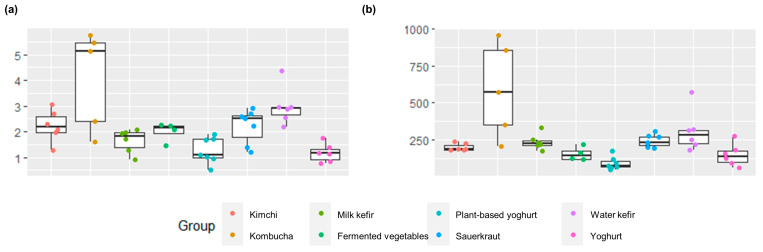
Box plots showing (**a**) Shannon’s alpha diversity index and (**b**) Chao richness estimates for each sample and group. There were significant differences for each estimate across group, namely, p_shannon_ = 0.00031, and p_chao1_ = 0.000094. The diversity and richness estimates for each sample can be found in Appendix A.

**Figure 2 foods-12-03827-f002:**
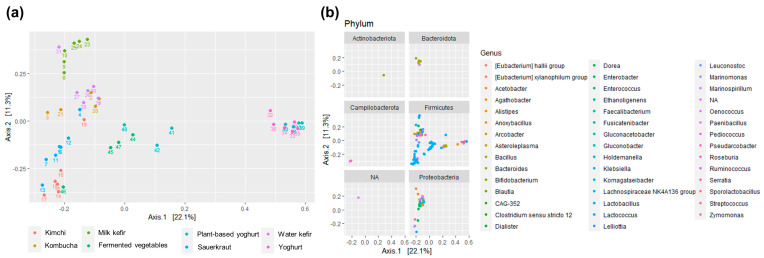
Dissimilarity in bacterial composition across foods and food groups (i.e., kimchi, kombucha, milk kefir, fermented vegetables, plant-based yoghurt alternatives, sauerkraut, water kefir, and yoghurt) visualized using PCoA based on Bray-Curtis distance and relative abundance ASV data. (**a**) shows the overall similarities and differences across samples; (**b**) indicates the prevalence and contribution of the different phyla and genera. For visualization purposes, ordination was limited to ASVs having an abundance of ≥1% in at least one sample, thus limiting the number of genera to 44 instead of the original 386. Abbreviations are as follows: ASV: amplicon sequence variants; PcoA: principal coordinate analysis. Notably, the database used to generate this figure used the misspelled name “Campilobacterota”; the correct name is Campylobacteriota.

**Figure 3 foods-12-03827-f003:**
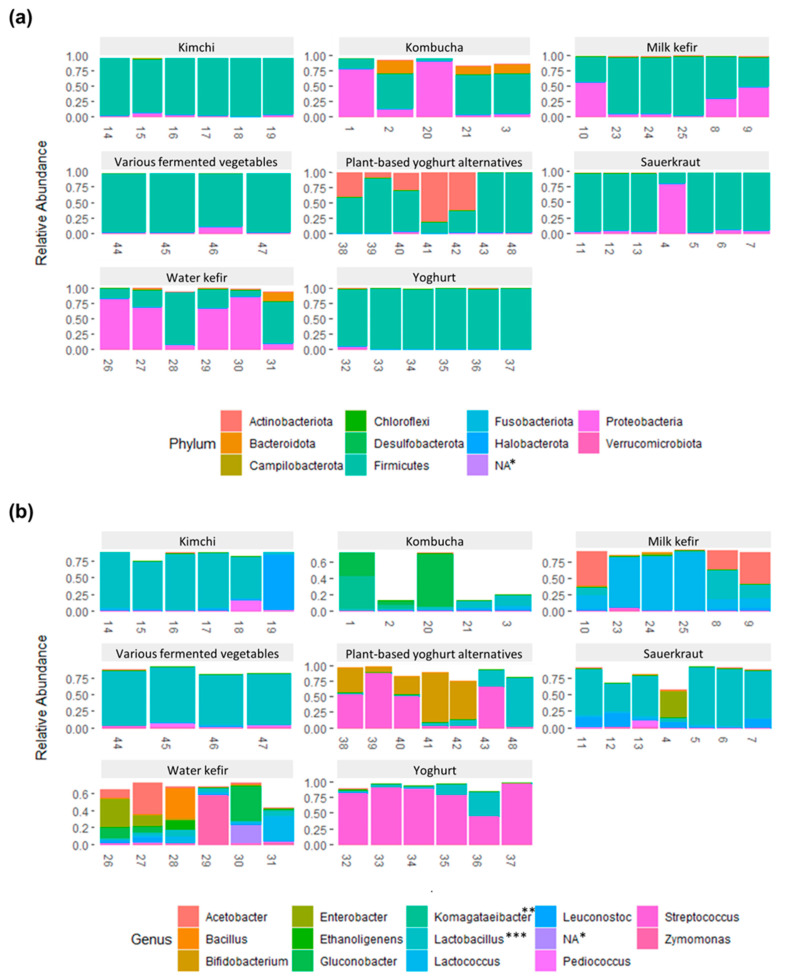
Bar plot showing the relative abundance of bacteria on (**a**) phylum level and (**b**) genus level. For visualization purposes, only amplicon sequence variants (ASVs) with an abundance of ≥0.1% for phyla and ≥10% for genera in at least one sample were included, which limited the number to 11 phyla and 14 genera (originally 33 phyla and 386 genera). Therefore, samples with higher proportions of rare and very rare taxa will have bars with a total abundance below 1.00 (100%). Taxonomy was based on assignments using the SILVA rRNA database. * Bacteria without phylum level match ** Matched as an unknown *Acetobacteraceae* species on ASV level *** Includes genera formally classified as *Lactobacillus*, i.e., *Lactiplantibacillus*, *Latilactobacillus*, *Lentilactobacillus*, *Levilactobacillus*, *Loigolactobacillus*, and *Schleiferilactobacillus*, as well as *Lactobacillus delbrueckii* group, *Paralactobacillus Holzapfelia*, *Amylolactobacillus*, *Bombilactobacillus*, *Companilactobacillus*, *Lapidilactobacillus*, *Agrilactobacillus*, *Lacticaseibacillus*, *Dellaglioa*, *Liquorilactobacillus*, *Ligilactobacillus*, *Furfurilactobacillus*, *Paucilactobacillus*, *Limosilactobacillus*, *Fructilactobacillus*, *Acetilactobacillus*, *Apilactobacillus*, and *Secundilactobacillus* [21].

**Figure 4 foods-12-03827-f004:**
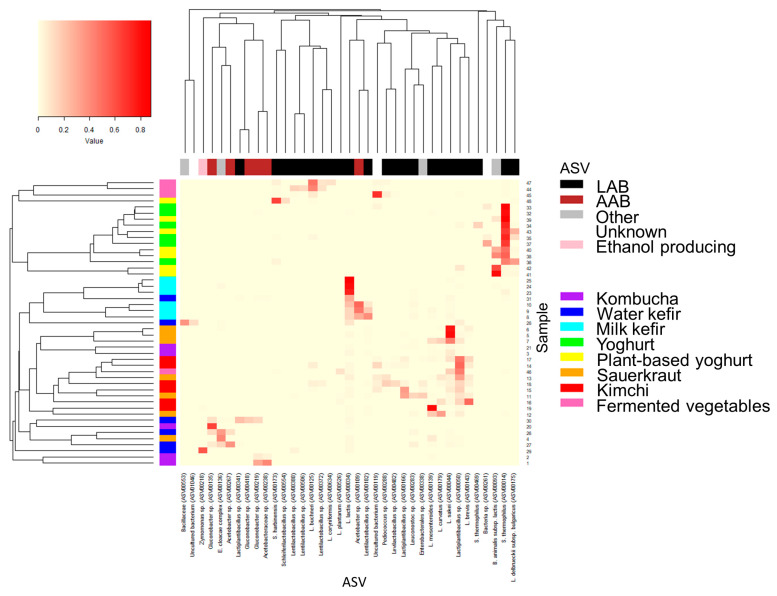
Heatmap of the most abundant bacteria for each fermented food. High (relative) abundance was defined as ASVs present at ≥10% in at least one fermented food. The relative proportion as well as the categorization of ASVs into LAB, AAB, ethanol-producing, or “other” (i.e., all other bacteria) are indicated in the colour legends in addition to the fermented food group for each sample. Hierarchical clustering was performed to group ASV and samples, respectively, based on similarities using the Bray–Curtis dissimilarity matrix and average linkage. Abbreviations are as follows: AAB: acetic acid bacteria; ASV: amplicon sequence variants; LAB: lactic acid bacteria.

**Table 1 foods-12-03827-t001:** Fermented foods and their respective fermented food groups, sample number, origin, and list of ingredients. Origin refers to either home environment, as indicated by individuals’ numbers, or commercial, as indicated by the name of the company.

Sample	Source	Ingredients
Kombucha		
20	Humm Kombucha	Water, raw sugar, black tea, and living kombucha culture
21	Renee Voltaire	Water, raw sugar cane, green tea (chun-mee), and kombucha culture
1	Individual 2	2 black teas, 1 oolong tea, sugar cane, and SCOBY (mid–late December)
2	Individual 2	2 black teas, 1 oolong tea, sugar cane and SCOBY (late December–mid January)
3	Individual 2	2 black teas, 1 oolong tea, sugar cane and SCOBY (mid–end of January)
Water Kefir		
28	Renee Voltaire	Water, agave, sugar, lemon juice, ginger, and living water kefir culture (*L. lactis lactis*, *L. lactis cremoris*, *L. lactis diacetylactis*, *L. casei*, *L. acidophilus*, and *Kluyveromyces marxianus* (yeast))
29	Grönt Levande	Water, raw sugar, water kefir grains, ginger, lime, and figs
30	Kombucheriet	Water, living water kefir culture, raw sugar, and rosehip
31	Kombucheriet	Water, living water kefir culture, raw sugar, and rhubarb juice
26	Individual 1	Water, muscovado sugar, and water kefir grains (3 days). Same batch of water kefir grains as Sample 27
27	Individual 1	Water, muscovado sugar, and water kefir grains (4 days). Same batch of water kefir grains as Sample 26
Milk Kefir		
23	Valio	Pasteurized milk, lactase enzyme, starter culture (bacteria and yeast), *L. rhamnosus* GG, and vitamin D (2,5% fat)
24	Arla	Pasteurized milk, starter culture, *L. acidophilus* LA-5, *B. lactis* BB-12, *L. casei* F-19 and vitamin D (3.0% fat)
25	Individual 1	Milk (Garant eko 3% fat) and starter culture from yo*gut; made on the same date as Sample 8
8	Individual 1	Milk (Garant eko 3% fat) and milk kefir grains from yo*gut (see legend for microbial composition*); same date as Sample 25
9	Individual 1	Milk (Garant eko 3% fat) and milk kefir grains from yo*gut (see legend for microbial composition*); milk kefir grains from same batch as Sample 10
10	Individual 1	Milk (Garant eko 3% fat) and milk kefir grains from yo*gut shop (see legend for microbial composition*); milk kefir grains from same batch as Sample 9
Yogurt (plain)		
32	Milchbauern Schrozberger	Milk (3.5%), skimmed milk powder, living bacteria culture, *L. acidophilus*, and *B. bifidum*
33	Wapnö	Pasteurized milk, cream, and yogurt culture (8% fat, total)
34	Larsa Foods	Homogenized and pasteurized milk (0%) and yogurt culture
35	Individual 1	Milk (Arla 3% fat) and bacterial culture blend from YOGUT ME
36	Individual 1	Milk (Arla 3% fat) and bacterial culture from Yogourmet (*L. delbrueckii* *bulgaricus*, *S. thermophilus*, *L. acidophilus*)
37	Individual 1	Milk (Arla 3% fat) and bacterial cultures from Cultures for Health
Plant-based yoghurt		
38	My Love My Life	Water, almond, tapioca starch, stabilizer, locust bean gum, salt, and lactic acid culture
39	So Soja	Soya drink (organic) 99% and live cultures including *Bifidobacterium* and *L. acidophilus*
40	Harvest Moon	Coconut milk 99%, tapioca, yogurt culture, *S. thermophilis*, *L. bulgaricus*, *L. acidophilus*, and *B. lactis*
41	Individual 1	Coconut milk, cashew nut, tapioca starch, and YOGUT ME culture blend
42	Individual 1	Coconut milk and YOGUT ME culture blend (batch 1)
48	Individual 1	Coconut milk and YOGUT ME culture blend (batch 2)
43	Individual 1	Coconut milk and Belle and Bella yoghurt culture
Sauerkraut		
11	Grönt Levande	White cabbage, onion, salt, garlic, juniper berry, and mustard seeds
12	Ölands Ljuvliga	White cabbage and salt
13	Tistelvind	White cabbage, cumin, juniper berry, and sea salt
5	Individual 1	White cabbage and salt (2%)
6	Individual 1	White cabbage, turmeric, garlic, and salt (2%)
7	Individual 1	Red cabbage and red onion
4	Individual 2	White cabbage, caraway, bay leaves, and salt (2.5%).
Kimchi		
14	Tistelvind	Savoy cabbage, carrot, daikon radish, onion, garlic, ginger, chili pepper, and sea salt (“hot”, jar 1)
15	Tistelvind	Savoy cabbage, carrot, daikon radish, onion, garlic, ginger, chili pepper, and sea salt (“hot”, jar 2)
17	Tistelvind	Savoy cabbage, carrot, daikon radish, onion, garlic, ginger, chili pepper, and sea salt (“medium hot”, jar 3)
18	Tistelvind	Savoy cabbage, carrot, daikon radish, onion, garlic, ginger, chili pepper, and sea salt (“medium hot”, jar 4)
16	Grönt Levande	White cabbage, onion, carrot, garlic, salt, radish, ginger, chili, water kefir, and bacterial culture
19	Individual 2	White cabbage, spring onion, ginger, garlic, kimchi chili powder flakes, and salt
Fermented vegetables		
44	Tistelvind	Beetroot, white cabbage and sea salt
45	Tistelvind	Carrot and sea salt
46	Grönt Levande	Cucumber and sea salt
47	Tistelvind	Cucumber and sea salt

* *L. lactis lactis*, *L. lactis cremoris*, *L. lactis lactis bv*. *diacetylactis*, *L. brevis*, *Leuconostoc* ssp., *S. thermophiles*, and *Kluiveromyces lactis* (yeast) and *Saccharomyces cerevisiae* (yeast).

## Data Availability

The data presented in this study is available in the Appendix A. Further inquiries can be directed to the corresponding author.

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
