# Peer review of "Characterization of the Bacterial Composition of 47 Fermented Foods in Sweden"

_foods, 2023, doi:10.3390/foods12203827_

Round 1

Reviewer 1 Report

The manuscript reports the characterization of the bacterial microbiota of different fermented foods from Sweden. The Authors analyzed 47 fermented foods belonging to different categories, including dairy and vegetable products, retailed or produced at home, characterizing their microbial composition through 16S rRNA gene sequencing. 

Although the topic is timely and sound, and the Authors performed a lot of work, collecting many data, I found several weak points that limit the impact of the article in its present form. I recommend an extensive revision.

The main issues are the following

•          The significance of the results is not clearly evidenced. The goal is not clear: concerning commercial products, they are obtained through standardized procedures and by using defined and selected starter cultures, usually composed of defined strains. What is the sense to analyze the microbial composition? Was the aim to verify if the microbial composition of the final foods reflects the composition of the starter cultures used for the production? Or was the aim to compare the microbial composition among commercial and home-made products? 

The Authors state “This study provides information on bacterial communities in a diverse range of fermented foods, contributing to the understanding of the microbiota's role in product traits and potential impacts on human health”: this aim was not achieved. The Authors do not discuss on these aspects, they only describe the results, and the conclusion is that the products have different microbial composition.

•          Overall, the discussion is poor and should be extensively improved.

•          My impression is a general lack of a microbiological perspective when presenting and describing the results.

•          The samples are not homogeneous: in some food groups there are replicates of the same brand or of home-made products, in some other groups not, and this makes the results difficult to compare.

•          When analyzing food microbiota composition, the information relative to fermentation conditions, such as the temperature, as well as to the ingredients, (for dairy products if the milk used was pasteurized or sterilized; for vegetable products the presence of salt, spices, etc.), is of utmost importance due to their possible impact on microbes, in order to evaluate the obtained results.

Minor points

•          The microbiological terminology should be more accurate: sp. or spp. should be indicated after the genus and not after taxa higher than the genus; sometimes the Authors refer to taxa but it would be more appropriate referring to genus; sometimes they refer to the generic term ASV and this is confounding 

•          The order of appearance of the fermented foods should be uniformed among the tables and figures 

Specific comments

Abstract: 

Lines 25-27: “This study provides information on bacterial communities in a diverse range of fermented foods, contributing to the understanding of the microbiota's role in product traits and potential impacts on human health”: this goal was not fulfilled. The Authors do not discuss on these aspects 

Introduction: 

Lines 52-57: why mentioning only Asian and African products? what about European countries? Especially Mediterranean countries (but also France, Ireland, etc.) produce a lot of traditional fermented foods and several published papers focus on their microbiota characterization (for example different cheeses, fermented meat products, table olives, to cite some of them) 

General comment: the Authors should also mention the differences among commercial and traditional fermented foods in terms of microbial composition, due to the use of commercial, defined starter cultures or of the autochthonous microbiota, respectively. 

Materials and Methods: 

More detailed information should be provided about the procedures applied for preparing foods in home environments. The temperature was not specified. Was it room temperature? Concerning yogurts, the fermentation temperature should be 42-45 °C.  

What does “different starter cultures” (line 96) mean? 

Concerning kefir, what were the differences among the bacterial composition of kefir grains and commercial starter cultures (lines 91-92)?  

In general, the Authors should have performed 16S rRNA gene sequencing also on starter cultures, kefir grains, SCOBY and so on, in order to give a better interpretation of the results obtained from the resulting fermented products, by comparing the initial and the final composition of microbiota. 

Lines 102-104: freezing without a cryoprotecting agent can compromise the vitality of microbes, resulting in cell lysis and consequent loss of nucleic acids, which in turn could alter the results of NGS analysis. How can the Authors exclude the loss of some microbial species? 

Lines 111-112: referring to 4 ml is not appropriate: what about solid foods? Were they homogenized prior to the analysis? More details are needed. Moreover, how many grams were analyzed? 

Lines 148-154: NGS is reliable to the genus level, while at the species level the accuracy cannot be assured. This aspect should be mentioned here.

Line 184: Table S3 reports “Lactobacillus” genus, without considering its recent re-classification into novel 25 genera. The Authors should indicate all the genera formerly reported as Lactobacillus, since these genera in many cases were useful to differentiate distinct products (this comment also applies to Figure 3b, see below)

Table 1:

Line 141: Replace “Origin” with “Source”

A column describing the fermentation conditions should be added

To my opinion, there is no need to maintain the excluded products in the Table, since it is confounding to the reader

Milk kefir: for commercial products, pasteurized milk is reported, while for the home-made products it is not specified if the used milk was pasteurized, UHT, etc.

It is not clear if products 25 and 8 are the same: what does it mean “same date of Sample 25 (or 8)? Were the starter cultures used different (“culture from yo*gut” and “milk kefir grains from yo*gut shop”)? What was the rationale for using different cultures?

Concerning Sample 24, is the starter culture composed exclusively of the indicated probiotic strains (L. acidophilus LA-5, B. lactis BB-12, L. casei F19)? If this is the case, how can this product be classified as kefir?

Yogurt: as for kefir, for commercial products (except one), pasteurized milk is reported, while for the home-made products it is not specified if the used milk was pasteurized, UHT, etc.

Concerning Sample 32, L. acidophilus and B. bifidum, reported as living bacterial cultures, do not represent the species required to make yogurt (namely L. delbrueckii subsp. bulgaricus and S. thermophilus). How can this product be classified as yogurt?

Plant-based yogurt: concerning Sample 38, what is B. species?

Results and Discussion:

In general, the discussion is poor throughout this section and should be extensively improved.

Lines 216-219: A reference to the sample numbers could help the reader to identify them in Figure 2a. Moreover, which is the significance of these results? This should be discussed in the text

Figure 2: panels a and b are not cited in the text. The terms Bacteroidetes/Bacteroidota should be uniformed in figure and text.

Line 228: please correct “Proteobacterium phyla” with “Proteobacteria phylum”

Line 230: please correct “Campilobacteriota” with “Campylobacteriota”

Line 231: the food and food groups should be specified. No discussion is provided. Are the less represented phyla expected in the analyzed foods? What about the potential pathogens/alteratives (Serratia, Klebsiella, Campylobacteriota)

Lines 243-249: no mention to the correspondence among the specific genera and the foods

Figure 3b: the Authors should indicate all the genera formerly reported as Lactobacillus, since these genera in many cases were useful to differentiate distinct products. 

Figure 4: this figure is very difficult to consult. The names of genera and species are don legible

Lines 268-272: LAB and AAB are very generic definitions, it would be preferable to mention the genera in relation to each specific food. For this reason, I would suggest to remove Figure 5 (or move it to supplementary material), since it does not appear to be so informative.

Paragraph 3.4. Bacterial composition and variability within each fermented food group: in general, throughout the paragraph, the Authors should refer also to Figure 3b, which describes the genera that are mentioned in the text. Moreover, a more in-depth discussion should be provided.

Lines 292-297: I would expect that the main differences are evident at the species or strain level, rather than at the genus level. Moreover, what about the composition of SCOBY? This information would greatly help in the interpretation of the results. Without knowing the initial composition of SCOBY or kombucha cultures used for preparing the foods the significance of the results is not so sound.

Lines 319-321: from Figure 3b, Sample 28 is dominated by Bacillus. This genus does not appear among the starter cultures reported in Table 1. Please, give an explanation.

Lines 326-354: as suggested before, what about the composition of starter cultures? This information would greatly help in the interpretation of the results. Also, the milk characteristics (sterilized, pasteurized), as well as the temperature of fermentation process (especially for yogurt) are key aspects to be taken into account.

Lines 350-354: please, comment on these findings. In particular, why L. delbrueckii subsp. bulgaricus was not found in some of the samples? It represents one of the two bacterial species employed to produce yogurt.

Lines 362-365: the resemblance between the two products is likely dependent on the starter culture used, rather than the ingredients. Was the microbiolgical composition of the starters known? As suggested, 16S rRNA gene sequencing should have been performed also on starters. 

Lines 366-367: please, provide an explanation for why you did not detect L. acidophilus.

Line 377: Please specify which Supplementary material.

Lines 421-444: Final remarks and conclusions are not exhaustive (see general comments above).

Reviewer 2 Report

This research follows a systematic approach, which holds significant importance for the development of local fermented food in Sweden. The experimental work is relatively straightforward, utilizing amplicon sequencing primarily to characterize the bacterial community. I would suggest that the author consider making some revisions based on the following suggestions:

1. Classification of Fermented Food: In my opinion, fermented foods can be categorized into three main groups: fermented tea, fermented vegetables, and fermented dairy products. Consequently, I recommend that the author provide a detailed classification based on this standard. The current classification makes it difficult to determine the hierarchical relationship among Sauerkraut, Kimchi, and fermented vegetables. It is advisable for the author to use local names to denote these fermented foods since detailed instructions on their formulation have already been given.

2. Taxonomic Annotation of Lactobacillus: It seems that the taxonomic annotation of Lactobacillus is only updated in the result analysis, while the charts obtained through bioinformatics analysis still employ outdated taxonomic names for lactic acid bacteria. As far as I know, the latest database can accomplish this objective. Therefore, I suggest that the author provide comments on the chart.

3. Analysis of Metabolic Function Prediction: I recommend that the author expand the analysis of metabolic function prediction to compare the functions of different types of fermented foods. Picrust2 can effectively achieve this task, and I believe this aspect of the research can enhance the value of the article.

4. Title: "bacterial microbiota" should be replaced by "bacterial community".

Good!

Round 2

Reviewer 1 Report

The Authors have satisfactorily addressed all my questions and the quality of the manuscript has been improved.

Author Response

Thank you again for your comments and their contribution towards improving the manuscript. We were happy to hear our revisions satisfactorily addressed you questions. 

Reviewer 2 Report

Although the author has responded to my comments, I have noticed a discrepancy between the taxonomic annotation results of Lactobacillus in Figures 2 and 3 and the corresponding text description. While I understood most of the author's explanations, I recommend updating Figures 2 and 3 to ensure consistency, as amplicon sequencing typically focuses on genus-level taxonomic classification for accuracy.

nothing

Author Response

Dear reviewer,

Regrettably, we cannot fully address your comment suggesting changing Figures 2 and 3 regarding the annotation of Lactobacillus. Updating the figures and the genus-level results would entail a complete re-analysis of the results and their interpretation, since the data used for the figures were generated using the SILVA database and the classification it used at the time of analysis. However, all bacteria present at >1% in at least one sample have been manually classified using NCBI blastn suite and the current taxonomic annotation. These bacteria make up 85-98% in most samples as shown in Tables S10-S17.  Furthermore, we have now updated Table S2 to include sequences for all 2497 ASVs, which allows for transparency and for readers to make further identifications. The title and description of this table reflect this change and now reads “Table S2. Relative abundance of ASVs per sample and each ASV sequence” and “The relative abundance of all ASV for each sample and the ASV sequence used for taxonomic annotation”. As in previous versions of the manuscript, we acknowledge this limitation on the genus level in the methods and the results sections and provide a reference for the updated taxonomic annotation. Although not entirely what you propose, we hope you can accept this revision.